# Commercial Optical and Acoustic Sensor Performances under Varying Turbidity, Illumination, and Target Distances

**DOI:** 10.3390/s23146575

**Published:** 2023-07-21

**Authors:** Fredrik Fogh Sørensen, Christian Mai, Ole Marius Olsen, Jesper Liniger, Simon Pedersen

**Affiliations:** AAU Energy, Aalborg University, Niels Bohrs Vej 8, 6700 Esbjerg, Denmark; ffso@energy.aau.dk (F.F.S.); chrimai@energy.aau.dk (C.M.); omo@energy.aau.dk (O.M.O.); jel@et.aau.dk (J.L.)

**Keywords:** sensor testing and evaluation, multiple-sensor systems, imaging sensors, acoustic sensors, sonar measurements, stereo vision, laser triangulation, illumination, turbidity

## Abstract

Acoustic and optical sensing modalities represent two of the primary sensing methods within underwater environments, and both have been researched extensively in previous works. Acoustic sensing is the premier method due to its high transmissivity in water and its relative immunity to environmental factors such as water clarity. Optical sensing is, however, valuable for many operational and inspection tasks and is readily understood by human operators. In this work, we quantify and compare the operational characteristics and environmental effects of turbidity and illumination on two commercial-off-the-shelf sensors and an additional augmented optical method, including: a high-frequency, forward-looking inspection sonar, a stereo camera with built-in stereo depth estimation, and color imaging, where a laser has been added for distance triangulation. The sensors have been compared in a controlled underwater environment with known target objects to ascertain quantitative operation performance, and it is shown that optical stereo depth estimation and laser triangulation operate satisfactorily at low and medium turbidites up to a distance of approximately one meter, with an error below 2 cm and 12 cm, respectively; acoustic measurements are almost completely unaffected up to two meters under high turbidity, with an error below 5 cm. Moreover, the stereo vision algorithm is slightly more robust than laser-line triangulation across turbidity and lighting conditions. Future work will concern the improvement of the stereo reconstruction and laser triangulation by algorithm enhancement and the fusion of the two sensing modalities.

## 1. Introduction

Just as in the case above water [1,2,3], a large variety of motivating applications and solution algorithms exist for the use of sensor information in many operational contexts, such as localization and inspection, including 2D/3D reconstruction of underwater objects and scenes [4]. Acoustic sensing is the premier sensing modality used in underwater environments due to the high speed of sound and low attenuation in water [5]. Simultaneously, many underwater sensing tasks such as inspection are advantageously performed using optical cameras because they deliver high sensing resolution and are easily interpreted by operators [6]. However, optical sensing is considerably affected by turbidity, attenuation, and lighting (both natural sunlight and artificial illumination), factors which do not significantly affect acoustic methods [7,8]. Hence, the sensing modalities have complementary advantages; combined sensing solutions lead to a robust solution which is often required for use in automated solutions, as noted in [9,10].

Given these complementary sensing effects, it is desirable to quantify the effects of environmental influences such as turbidity on sensing performance to elucidate the operational limitations for each sensing modality.

The contribution of this work is to reproduce and expand on previous works concerning the effect of environmental turbidity and lighting on target reconstruction by the precise control of target distance using a 3D servo-driven gantry; the recording of simultaneous stereo, color image, laser-triangulation, and acoustic imaging in the same controlled experiment; and quantitative evaluation of sensor noise and accuracy by conversion to real-world-unit point clouds for each sensor.

The hypotheses are that optical sensing accuracy will be negatively affected as a function of increasing turbidity; that an optimum illumination level that provides the best performance exists; and that it will break down when exceeding a certain turbidity and target distance; contrarily, the acoustic sensor should be negligibly affected by these environmental parameters.

The rest of the paper is organized as follows: firstly, related works are outlined; secondly, the materials and methods applied in the experiments are described, including the chosen commercial sensors and experimental facility; thirdly, the results from the sensor’s raw measurements are evaluated for their operating limits and accuracy, with examples of measurements additionally illustrated; finally, the discussion summarizes the qualitative and quantitative behavior of the sensors.

## 2. Related Work

Previous investigations have focused on different objectives: for example, the reconstruction of undistorted and clear visual images from subsea images for the purposes of presentation to operators and the reconstruction of 2D/3D objects for the purposes of object detection, segmentation, classification, and structural damage detection [4] have been studied. Both qualitative and quantitative investigations of this nature have been performed in recent years.

In O’Byrne et al. [11], an image repository was created with various target objects under varying turbidites using a setup with two waterproof cameras to test stereo reconstruction algorithms. Some algorithms for 3D reconstruction and damage detection were demonstrated on this dataset in O’Byrne et al. [12,13,14]. Just as the case above water, structured light can be added to the scene to aid in reconstruction, demonstrated by Aykin et al. [15], Bruno et al. [16].

In Mai et al. [17], the fidelity was evaluated for high-frequency sonar, stereo vision, and time-of-flight (ToF) cameras of determining distance to and shape of a target object, with a focus on the comparison of sensor accuracy and noise. It was shown that stereo vision delivers the highest measurement fidelity, followed by the ToF camera; finally, sonar has the lowest measurement fidelity. A ToF camera was also investigated in Risholm et al. [18], wherein the camera used a range-gated strategy to successfully reduce backscatter from turbidity, in this case, to monitor fish in turbid environments.

An example of using optical and acoustic sensing modalities together is shown in Roman et al. [19], where a high-frequency sonar, stereo imaging, and a laser triangulation method were compared for archeological 3D measurements in the Aegean Sea. It was shown that the sensing modalities all provide useable fidelity in the given environment; however, turbidity and other environmental influences were not measured. In Yang et al. [20], the emphasis was on examining sharpness and color reproduction under varying turbidity and lighting conditions using a monocular color camera. A ColorChecker and SFR chart were used to estimate the image quality and color reproduction.

More recently, in Scott and Marburg [21], the quantitative effects of turbidity on various stereo reconstruction methods showed that stereo vision depth estimation is possible with usable robustness under low (17NTU) and medium (20NTU) turbidity conditions. Apart from inspection tasks, visual sensors can also be used for concurrent localization, such as those described in Concha et al. [22], where localization and dense mapping are demonstrated from a monocular camera sequence.

## 3. Materials and Methods

To perform the experiments, a commercial sensor was selected to embody each of the sensing modalities; then, these sensors were mounted in a rigid aluminum frame to fix the extrinsics between the sensors themselves and the target objects. First, we describe the selected sensors and their specifications; then, we describe the experimental setup, including the data acquisition and the selected target objects used in the performance evaluation; and finally, we describe the experimental procedure.

### 3.1. Sensors

For each sensing modality, a commercial-off-the-shelf (COTS) sensor was selected based on the maximum sensing distance which was used in the experiments, 2 m, while maintaining a high sensing fidelity under the given distance range. The stereo and color camera modalities were both embodied by the Intel D435i camera [23], and the acoustic modality was embodied by the BluePrint subsea M3000d sonar [24].

#### 3.1.1. Stereo and Color Camera

A COTS stereo camera, the Intel D435i [23], embodied the optical sensing modality. This camera was chosen based on having a minimum 2 megapixel resolution color imager as well as on-board stereo imaging; in particular, it had built-in stereo depth estimation processing (to reduce the need for external computation in an end-use application). The stereo camera sensor specifications are given in Table 1. For the Intel D435i, the color imaging sensor was the OmniVision OV2740, while the stereo imaging sensors were OmniVision OV9282s. Since the stereo depth estimation is a built-in function of the camera, the main stereo-sensing specifications are listed in Table 2.

#### 3.1.2. Forward-Looking Imaging Sonar

The acoustic sensing modality was similarly embodied by a COTS forward-looking imaging sonar, the BluePrint subsea M3000d [24]; this sonar was selected for its small range resolution <1 cm and small angular resolution <1° with a suitable minimum distance of ≤0.1 m and a maximum distance of ≥1 m. The forward-looking imaging sonar sensing specifications are given in Table 3. For the purposes of this work, the sonar was used exclusively in the high-frequency mode shown on the right.

#### 3.1.3. Laser-Line Augmentation

The laser specifications are given in Table 4. The laser was fitted with a line-generation lens immediately after the focusing lens and was mounted in a waterproof enclosure with a flat port acrylic window. The laser was focused at approximately 2 m and was mounted to be within view of the color camera at both the minimum and maximum test distances. The closest observable distance was determined by the intersection of the laser plane with the lower plane of the camera field-of-view (FOV), and the maximum distance was determined by the intersection with the upper plane of the FOV, as shown in Figure 1.

#### 3.1.4. Turbidity Sensor

The turbidity sensor was an optical nephelometric sensor, model Aanderaa Turbidity Sensor 4296 [25]. The sensor was mounted to measure the turbidity in the forward direction towards the target into an unoccluded volume to avoid reflections from the pool’s interior surfaces and the water surface. The turbidity sensor’s main specifications are given in Table 5.

### 3.2. Experimental Setup

The experimental setup consisted of three overall parts: the sensors being tested, the test pool filled with test medium, and the test targets mounted on a 3D gantry (traverse). To ensure the extrinsics were fixed between the sensors, they were mounted on a rigid frame made of aluminum profiles, shown in Figure 2.

The test pool was filled with tap water, and Kaolin [26] was used to control the turbidity. The test targets were mounted on a 3D gantry which allowed them to be moved with respect to the sensor frame, such that the distance between the target and the sensor reference planes could be varied. The complete experimental setup is shown in Figure 3. To prevent disturbances from external light sources, the experiments were conducted in a laser-safety-rated laboratory where external lighting could be reduced to near zero levels.

### 3.3. Target Objects

Two target objects were used during the experimental measurements: an ISO 12233:2017 edge spatial frequency response chart (eSFR chart) [27] used for image quality analysis—this eSFR target was printed in a 16:9 format and was printed with near-infrared and visible reflective inkjet technology—and a metal cylinder that resembles part of an offshore structure. The eSFR chart was glued to an aluminum sandwich backing plate and is shown in Figure 4a; the metal cylinder is shown in Figure 4b.

### 3.4. Data Acquisition

The data acquisition was performed using the Robot Operating System (ROS) Noetic built on Ubuntu 20.04, running on an NVIDIA Jetson Xavier NX, which is located within the stereo camera submersible enclosure. The Xavier NX was connected through serial communication (RS232) to the turbidity and conductivity sensors, by USB 3.1 to the stereo camera, and by gigabit ethernet to a switch outside the experimental tank. The Xavier NX and sensors were powered using power-over-ethernet (PoE) from the switch, apart from the sonar, which had a separate power supply and ethernet connection. The interconnection between the sensor components and the data capture equipment are shown in brief in Figure 5. See also Figure 2 for the physical layout of the sensors.

### 3.5. Experimental Parameters

The experiments were conducted at a set of turbidities, target distances, and illumination settings. Table 6a lists the desired and achieved turbidities for the experiment series, including the standard deviation as given by fluctuations in the turbidity sensor measurement. Table 6b lists the desired and achieved distances for the experiment series, including the measurement uncertainty; note that when transitioning to/from the far distances, the sensor frame was moved within the pool and the target distance was re-initialized using an external laser distance meter. The used lighting levels are shown in Table 6c.

### 3.6. Experimental Procedure

The experiments were conducted using a repetitive procedure which is also illustrated in Figure 6. The inner loop corresponds to light level variations; the intermediate loop corresponds to distance variations; and the outer loop corresponds to turbidity variations. The procedure was designed to have the least experimental disturbances during variable changes since light changes cause no physical movements, whereas the control of Kaolin content is additive in nature.
The sensor frame is placed within the test pool.The test target is reset, and the base distance is measured using a laser distance meter.Measurements are performed at each distance:
(a)Measurements are performed at each light level:
i.Light level is set at the selected percentage: see Table 6cii.The experiment is allowed to settle for 10 s.iii.Sensor data are recorded in ROSbag format; then, point (i) is repeated.(b)The distance is changed by control of the gantry: see Table 6b; then, point (a) is repeatedKaolin is added until the desired turbidity is reached—see Table 6a—as measured by the turbidity sensor; then, point 3 is repeated.

## 4. Results

Using the ROSbags generated through the experiments, the performance of three sensing modalities has been evaluated: stereo depth estimation based on the built-in algorithm of the Intel camera—see Section A.3; laser triangulation implemented through the color camera and the MATLAB triangulation algorithm—see Section A.1; and the high-frequency imaging sonar—see Section A.2. For all modalities, the measurement accuracy has been analyzed through MATLAB, as described in the Appendix A.

### 4.1. Illumination Effects

The light level naturally influences the results for the optical methods, influencing both stereo depth estimation and laser-line triangulation. By review of the sensor measurements, it is evident that for both visual methods, the optimal light level in the experiments is 50%, with an example illustrated in Figure 7b. Less illumination, 25%, results in less clear features for stereo estimation and increased laser glare, shown in Figure 7a, while illumination levels of 75% to 100%, shown in Figure 7c,d, results in reduced contrast for the laser as well as increased backscatter, which reduces visual features in the resulting images.

### 4.2. Stereo Depth Estimation

For the stereo camera, the performance has been evaluated for a rectangular region of interest (ROI) in the central 20% of the depth image frame, as illustrated in Figure 8. To determine the operational limits, the cut-off for valid distance measurements has been set at 50% valid pixels within the ROI, i.e., a pixel fill rate of >50% is considered as valid. The measurement accuracy as analyzed with Section A.3 is shown in Table 7 and Figure 9, Figure 10 and Figure 11 while an example of the depth image is illustrated in Figure 8. Note how the background of the pool is still estimated at 0.3 FTU, Figure 8a, but begins to disappear at 2.1 FTU, Figure 8b, while the target remains valid in both cases. For the cylinder geometry estimation, the results show a very high deviation, which most likely stems from an insufficient quality of stereo intrinsic calibration, since it is evident that the eSFR plate behind the cylinder is also heavily distorted, as illustrated in Figure 12.

### 4.3. Laser Triangulation

The laser triangulation is performed by detecting the laser-line and projection as described in Section A.1, with examples shown on Figure 13 and results shown in Table 8 and Figure 14, Figure 15 and Figure 16. The laser triangulation has an accuracy of a single centimeter up to a range of about 50 cm, increasing to an error of 3 cm at a range of 200 cm. The behavior of the deviation over distance seems to indicate some remaining uncompensated error in the camera intrinsics calibration since the error is non-monotonic with respect to the target distance. The sensing functions up to a distance of 103 cm for turbidities of ≤2.1 FTU—see Figure 14 and Figure 15—and drops to 43 cm at 6 FTU: only very close range sensing is possible at this high turbidity; see Figure 13c and Figure 16. The laser-line is naturally much easier to detect due to the improved contrast at low turbidities, which is evident from Figure 13a,b. For the cylindrical target, the geometric reproduction accuracy is shown in Figure 17, where the detected circle has a radius close to the actual value of 5 cm; the main outliers stem from the specular reflection along the long axis of the cylinder. The deviation is increased as the distance to the cylinder target is increased, as shown in Figure 17c. Overall, the fidelity of the geometric reproduction is satisfactory at close distances.

### 4.4. Acoustic (Sonar)

The sonar data have been processed using the program described in Section A.2, with examples shown on Figure 18 and results summarized in Table 9 and Figure 19, Figure 20 and Figure 21. The sonar target object distances show excellent linearity >0.98% and consistent monotonic error for all turbidities. Of particular note in the resulting images is the specular acoustic artifact arising at close distances, which creates a radial high-intensity echo tangential to the plane of the target object. The cylindrical target information is illustrated with a binarized image in Figure 22, where it is clear that the cylinder is detected; however, there is a substantial amount of noise at the front and rear boundaries of the cylinder.

## 5. Discussion

In general, acoustic sensing is mostly stable across operating conditions; however, stereo vision and laser-line triangulation can also operate successfully under low and medium turbidity conditions: 0.3 FTU to 2.1 FTU at ranges of up to 100 cm. For laser triangulation, the accuracy is relatively constant in the range of 0.3 FTU to 2.1 FTU, with a total maximum mode deviation of 1.54 (0.90) cm at a range of 103 cm. Stereo depth estimation suffers from some non-linearity and increased deviation up to 36 cm—though, it is lower, ≤15 cm, at a distance below 140 cm; this most likely related to an insufficient quality of the intrinsic parameter calibration in particular, which warrants further work. At 6 FTU, the operating range is severely limited for the optical methods (laser-line and stereo depth estimation) but still usable up to distances ≤43 cm for laser distance measurements and up to ≤63 cm for stereo depth estimations. For all of the sensors, it is possible to detect and estimate the cylinder targets’ geometry within 10% of the actual dimensions at distances closer than 63 cm. However, accuracy is substantially worse at longer distances. The operating depth for the considered approaches is generally limited by the manufacturer constraints for the commercial sensors as noted in their specifications; for the laser triangulation, the operating depth is additionally limited by the amount of ambient light; operating very close to the surface would not be possible. For the stereo camera, a large distortion still remains after the execution of the built-in calibration procedures: this would most likely need to be further corrected in a real application, depending on the particular application requirements. For acoustic sensing, other environmental parameters such as salinity or suspended particulate matter of large sizes may be more interesting to investigate since these are more likely to affect performance and operating limits with respect to the target distance. In summary, this work entails that these optical methods are usable even under relatively high turbidities if they are used for operations where only short-range measurements are needed; the useful operating range increases with decreasing turbidites, up until a maximum experimental distance of 200 cm. Contrarily, ranging using the acoustic sensor is, for the purpose of detecting used target objects under the given distances and environmental effects, unaffected, even at the highest turbidity and target distance.

## 6. Concluding Remarks

The experimental evaluation confirms the hypotheses that these optical methods provide great spatial details of the target objects and that increased turbidity affects their accuracy negatively. However, even at some substantial turbidity levels, i.e., 2.1 FTU, they still provide reasonable target object information at close ranges. Conversely, the sonar is not affected to a notable degree by turbidities of up to 6 FTU, but it provides the least amount of spatial information. In summary, this warrants investigation of sensor fusion where the complementary advantages of the different modalities can be fully exploited. Other future work includes the possible improvement of the laser-line distance measurement algorithm to improve the operating range and rejection of specular reflections. Alternatively, a modulated or rotational laser approach can also be investigated. Improvement of the stereo camera calibration to lower the distortion or external processing of the stereo camera information can be studied in order to ascertain whether larger operating conditions are achievable with other algorithms; this can be performed in extension to or completely replace the built-in stereo depth estimation. The addition of other environmental influences, such as salinity and suspended particulate matter, may lead to additional effects worth investigating, particularly for acoustic sensing.

## Figures and Tables

**Figure 1 sensors-23-06575-f001:**
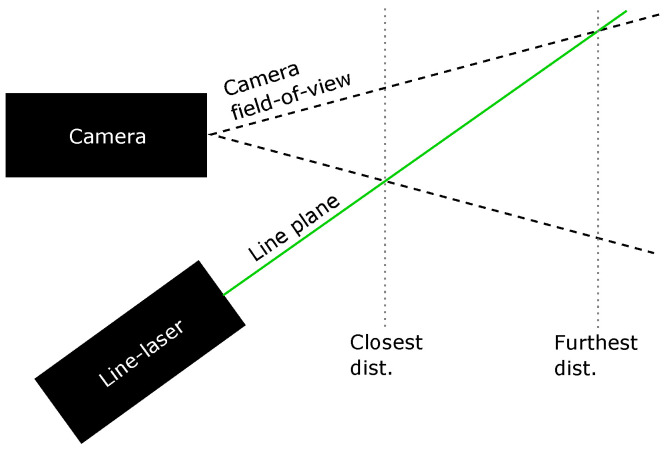
Laser (green) and camera field-of-view geometry (dashed lines).

**Figure 2 sensors-23-06575-f002:**
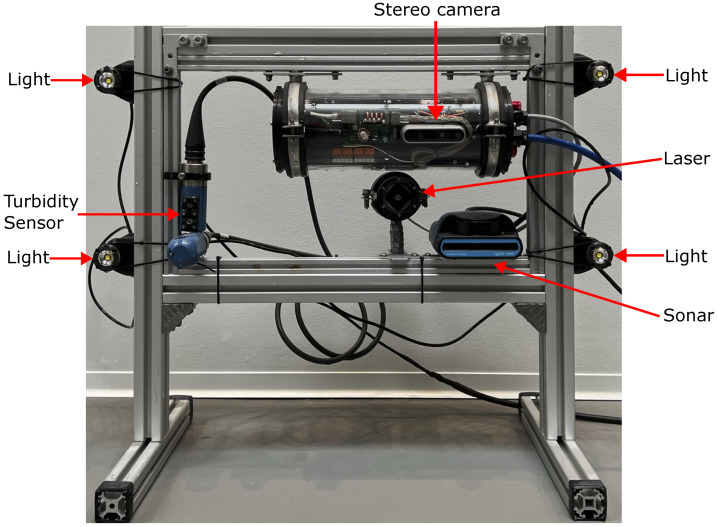
Image of the sensor frame. Upper right: stereo camera, lower right: sonar, center: laser, lower left: turbidity sensor; exterior of frame: 4 pcs. LED lights.

**Figure 3 sensors-23-06575-f003:**
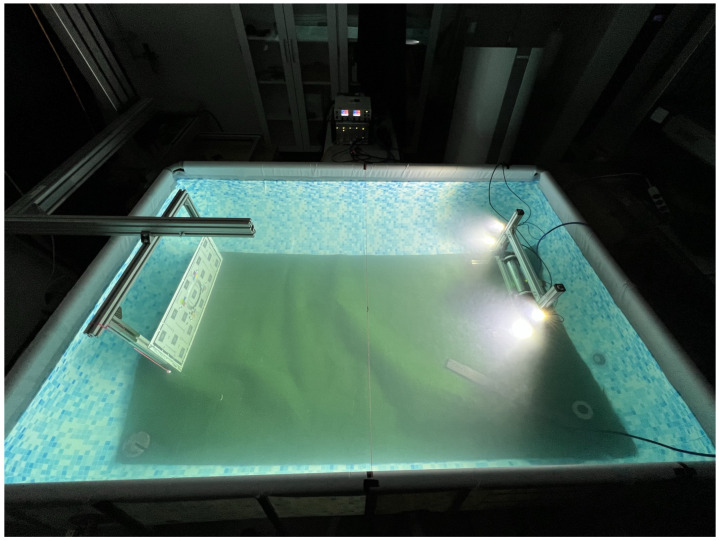
Image of the experimental setup. Left: target object on the gantry. Right: sensor frame with lights. Lower right: mixer. Pool dimensions: 180 cm by 260 cm.

**Figure 4 sensors-23-06575-f004:**
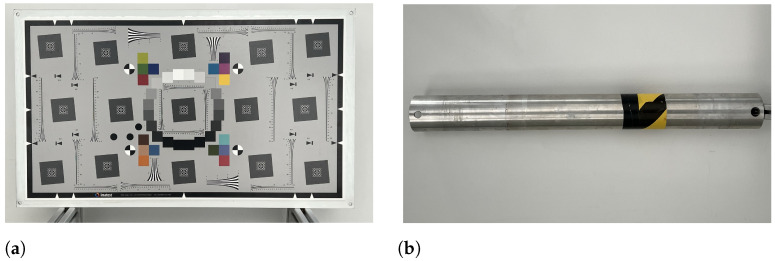
Pictures of target objects, eSFR, and metal cylinder. (**a**) eSFR target object [27]. (**b**) Cylinder target object (vertical during experiments). Material: aluminium, diameter: 10 cm, wall thickness: 5 mm.

**Figure 5 sensors-23-06575-f005:**
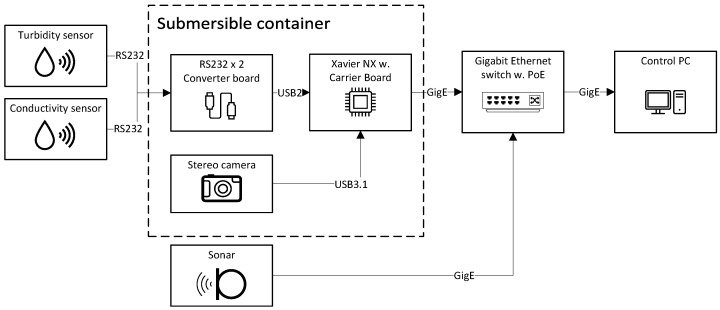
Block diagram of the data acquisition setup. The inner box denotes components in the transparent submersible container, wherein the stereo camera is mounted.

**Figure 6 sensors-23-06575-f006:**
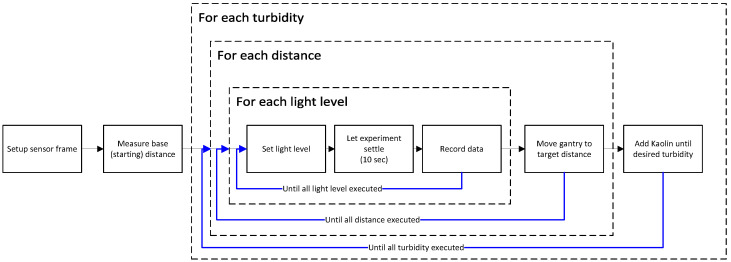
Block diagram of the experimental procedure. Black lines denote experimental flow, blue lines denote experimental repetitions.

**Figure 7 sensors-23-06575-f007:**
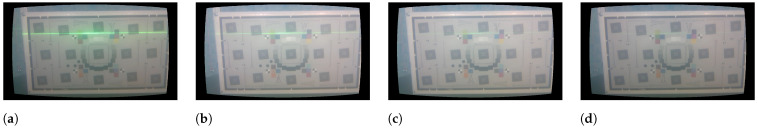
Color images of eSFR target at 1.03 m under varying light levels. (**a**) RGB image at light level of 25%, (**b**) RGB image at light level of 50%, (**c**) RGB image at light level of 75%, (**d**) RGB image at light level of 100%.

**Figure 8 sensors-23-06575-f008:**
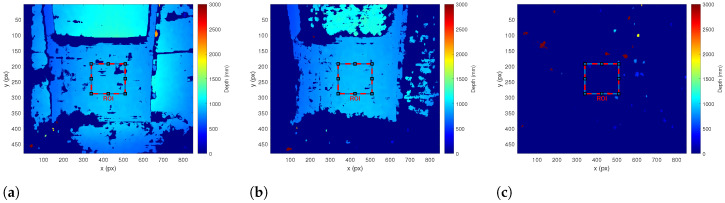
Depth images of eSFR target at 1.03 m. (**a**) Depth image of eSFR target at 0.3 FTU, (**b**) Depth image of eSFR target at 2.1 FTU, (**c**) Depth image of eSFR target at 6.0 FTU.

**Figure 9 sensors-23-06575-f009:**
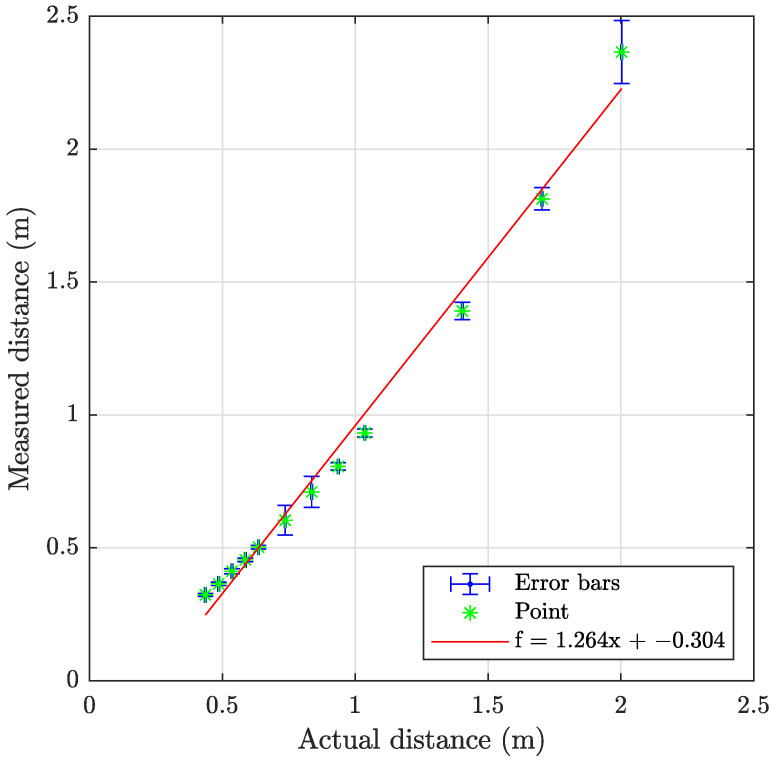
Graph showing the measuring accuracy at 0.3 FTU for stereo depth estimation.

**Figure 10 sensors-23-06575-f010:**
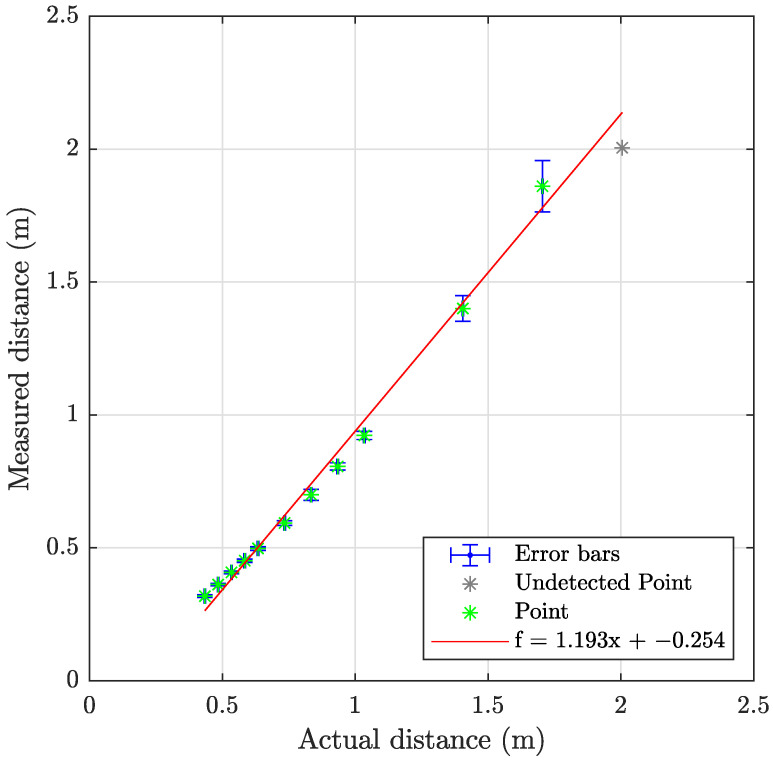
Graph showing the measuring accuracy at 2.1 FTU for imaging sonar.

**Figure 11 sensors-23-06575-f011:**
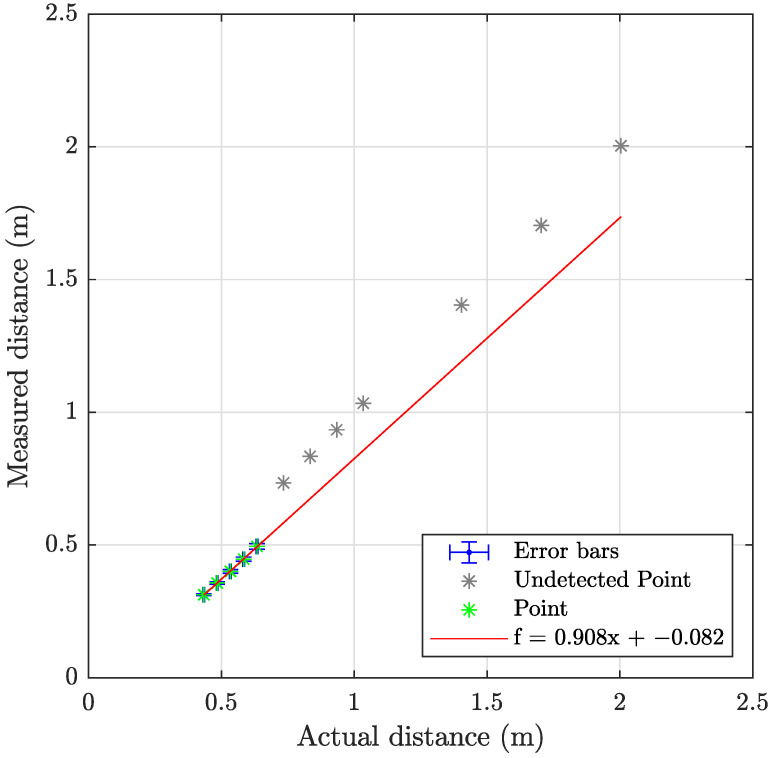
Graph showing the measuring accuracy at 6.0 FTU for imaging sonar.

**Figure 12 sensors-23-06575-f012:**
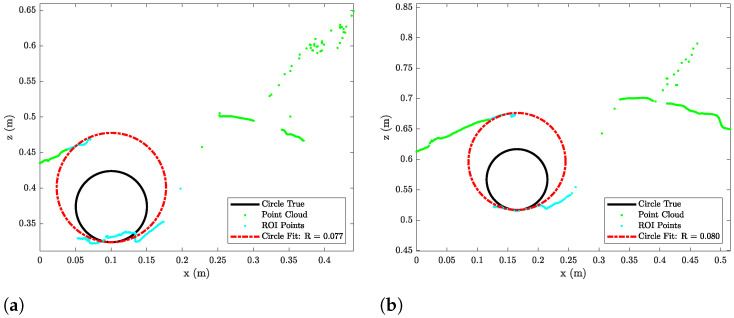
Circlefits of cylinder target at 0.3 FTU. (**a**) Circlefit of cylinder target at 0.64 m to eSFR target. (**b**) Circlefit of cylinder target at 0.84 m to eSFR target.

**Figure 13 sensors-23-06575-f013:**
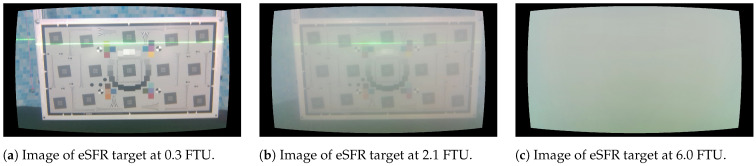
Images of eSFR target at 1.03 m.

**Figure 14 sensors-23-06575-f014:**
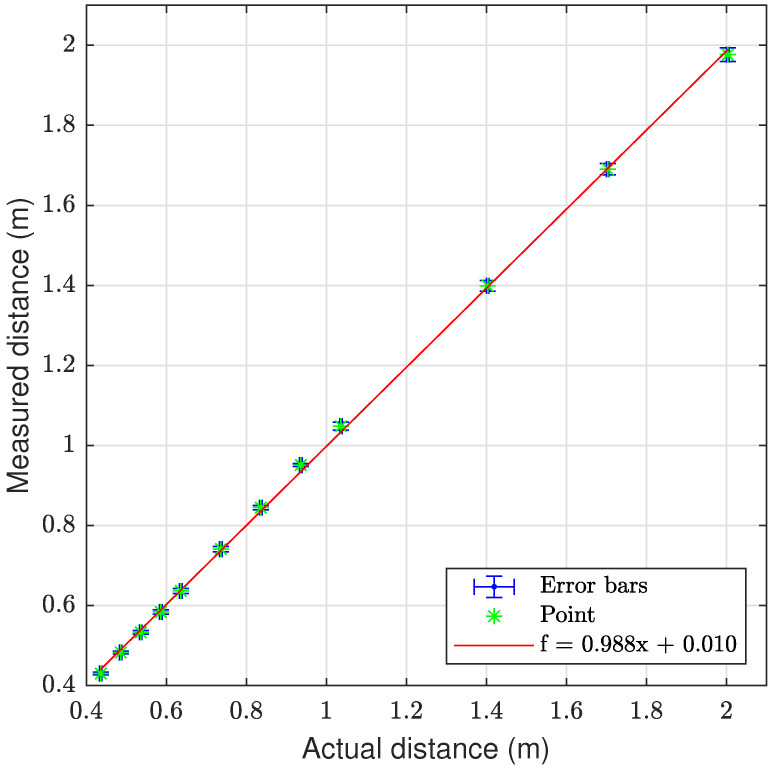
Graph showing the measuring accuracy at 0.3 FTU for laser triangulation.

**Figure 15 sensors-23-06575-f015:**
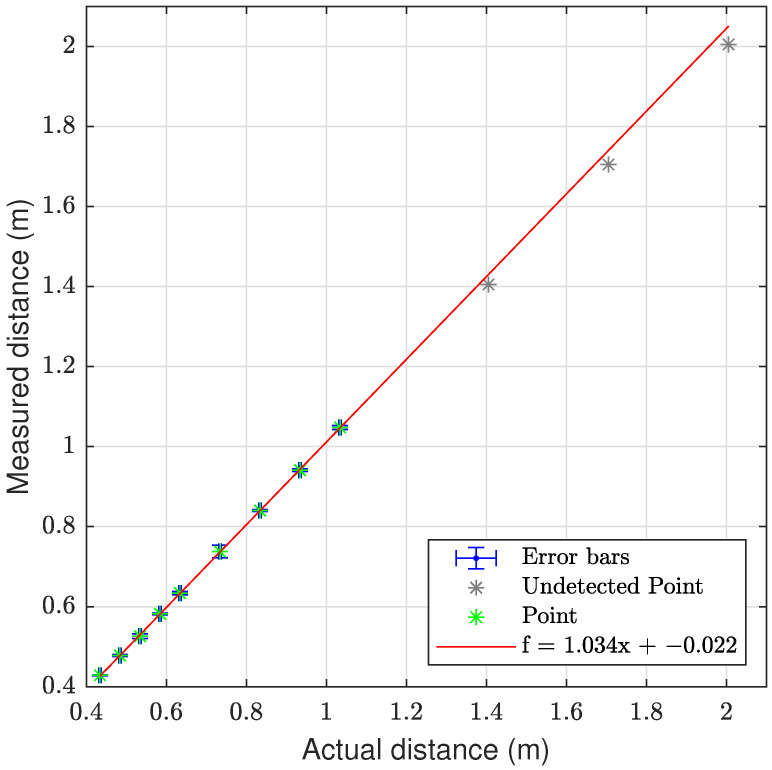
Graph showing the measuring accuracy at 2.1 FTU for laser triangulation.

**Figure 16 sensors-23-06575-f016:**
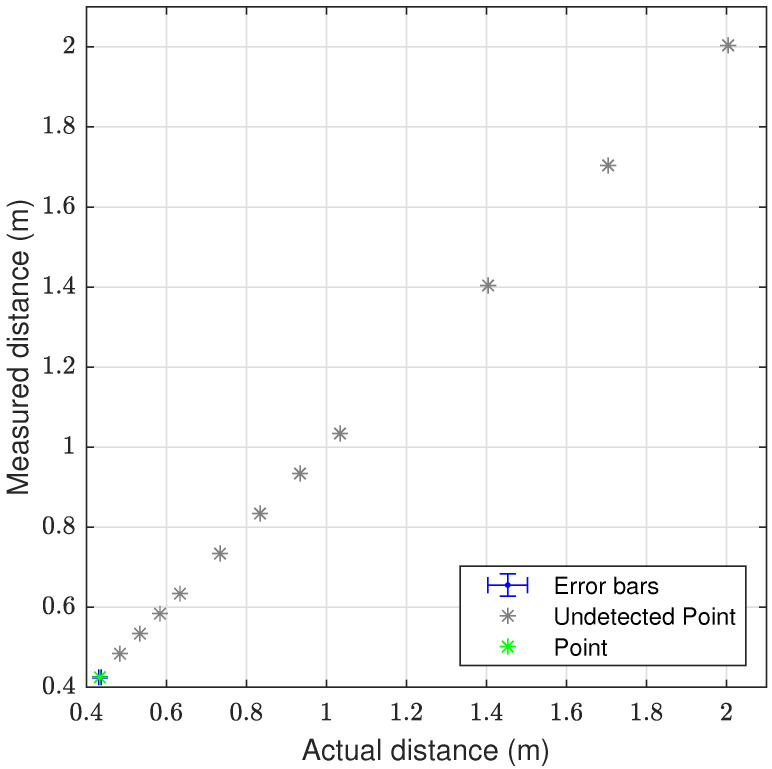
Graph showing the measuring accuracy at 6.0 FTU for laser triangulation.

**Figure 17 sensors-23-06575-f017:**
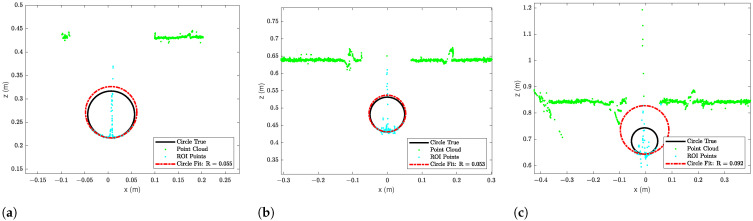
Circlefits of cylinder target at 0.3 FTU. (**a**) Circlefit of cylinder target at 0.44 m to eSFR target. (**b**) Circlefit of cylinder target at 0.64 m to eSFR target. (**c**) Circlefit of cylinder target at 0.84 m to eSFR target.

**Figure 18 sensors-23-06575-f018:**
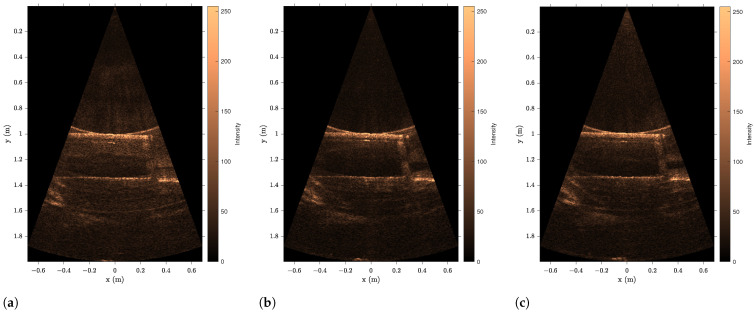
Sonar images of eSFR target at 1.03 m. (**a**) Sonar image of eSFR target at 0.3 FTU. (**b**) Sonar image of eSFR target at 2.1 FTU. (**c**) Sonar image of eSFR target at 6.0 FTU.

**Figure 19 sensors-23-06575-f019:**
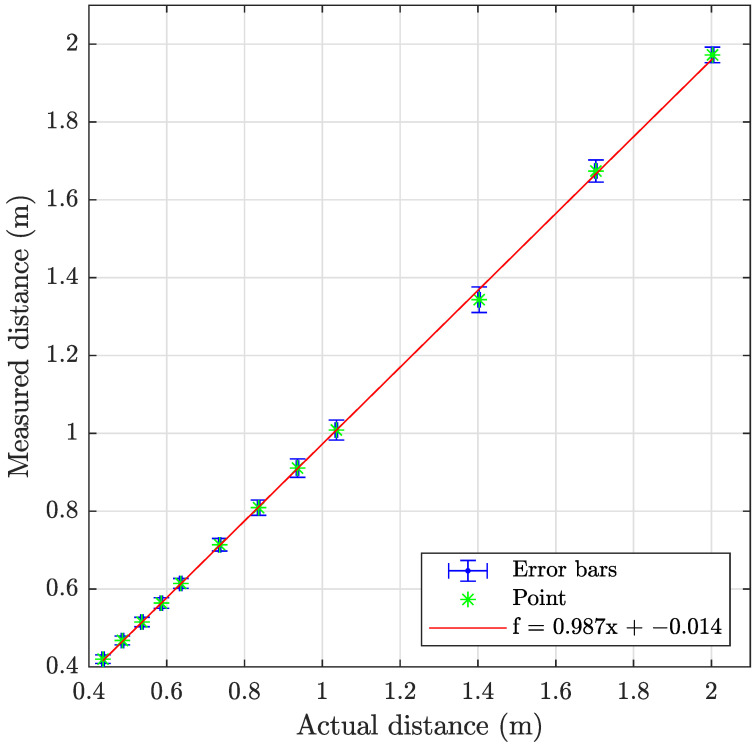
Graph showing the measuring accuracy at 0.3 FTU for imaging sonar.

**Figure 20 sensors-23-06575-f020:**
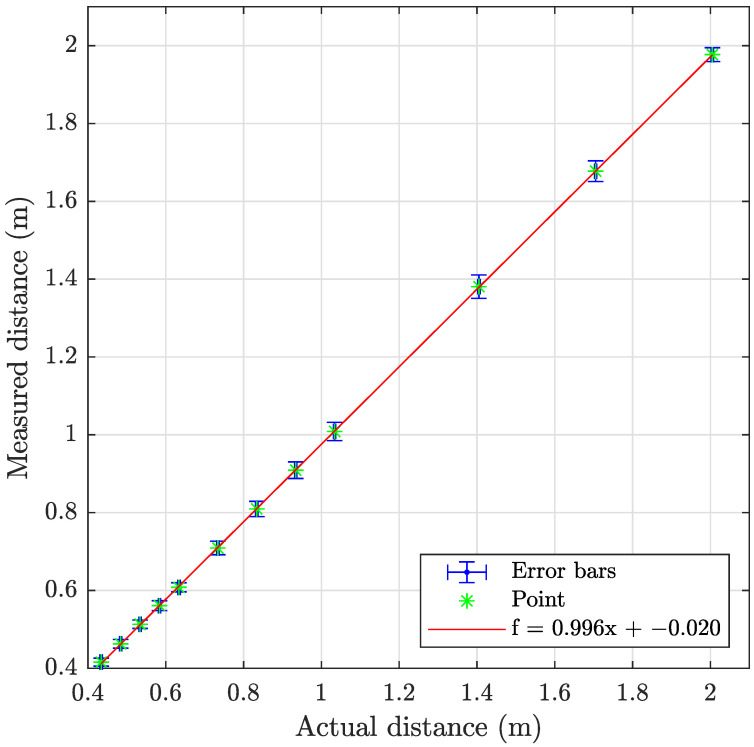
Graph showing the measuring accuracy at 2.1 FTU for imaging sonar.

**Figure 21 sensors-23-06575-f021:**
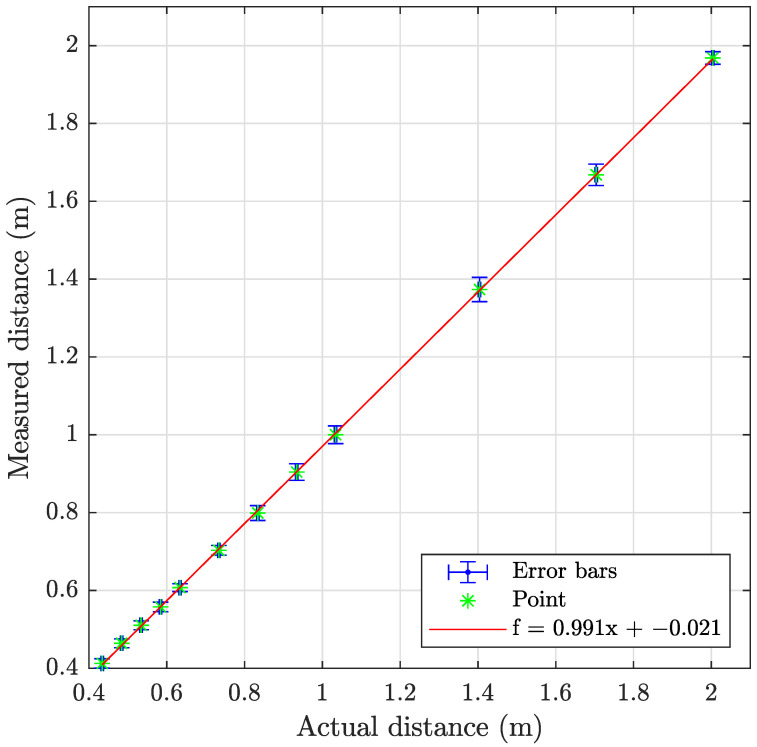
Graph showing the measuring accuracy at 6.0 FTU for imaging sonar.

**Figure 22 sensors-23-06575-f022:**
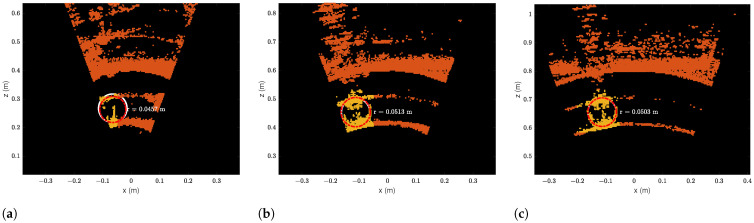
Circlefit of cylinder target at 0.3 FTU. (**a**) Circlefit of cylinder target at 0.44 m to eSFR target. (**b**) Circlefit of cylinder target at 0.64 m to eSFR target. (**c**) Circlefit of cylinder target at 0.84 m to eSFR target.

**Table 1 sensors-23-06575-t001:** Intel D435i imaging sensor manufacturer specifications in air.

Parameter	Stereo Imager	Color Imager
Resolution	1280 px × 800 px	1920 px × 1080 px
Shutter type	Global shutter	Rolling shutter
Data format	10-bit RAW	10-bit RAW RGB
Horizontal FOV	91 ± 1°	69 ± 1°
Vertical FOV	66 ± 1°	42 ± 1°
Diagonal FOV	101 ± 1°	77 ± 1°

**Table 2 sensors-23-06575-t002:** Intel D435i stereo depth estimation manufacturer specifications at a 2 m distance, recommended settings, in air.

Parameter	Value
Resolution	848 px × 480 px
Frame rate (max)	90 FPS
Data format	16-bit (1 mm/LSB)
Horizontal FOV	86 ± 3°
Vertical FOV	57 ± 3°
Diagonal FOV	94 ± 3°
Min. distance	195 mm
Depth accuracy	≤2%
RMS error	≤2%
Temporal noise	≤1%
Fill rate	≥99%

**Table 3 sensors-23-06575-t003:** Oculus m3000d sonar manufacturer specifications, * indicates range-dependent specification.

Common parameters		
Update rate max *	40 Hz	
Number of beams (max)	512	
Range (min)	0.1 m	
Vertical aperture	20°	
**Mode parameters**	**Low-frequency mode**	**High-frequency mode**
Operating frequency	1.2 MHz	3.0 MHz
Range (max)	30 m	5 m
Range resolution *	2.5 mm	2 mm
Horizontal aperture	130°	40°
Angular resolution	0.6°	0.4°
Beam separation	0.25°	0.1°

**Table 4 sensors-23-06575-t004:** Laser-line specifications.

Parameter	Value
Laser type	Diode laser
Laser wavelength	532 nm
Beam class	Class 3B

**Table 5 sensors-23-06575-t005:** Aanderaa Turbidity Sensor 4296 [25].

Parameter	Value
Range	0 FTU to 25 FTU
Resolution	0.1%
Accuracy	±3% of range

**Table 6 sensors-23-06575-t006:** Experimental conditions.

(**a**) Experimental turbidities.	
**Desired Turbidity**	**Average Measured Turbidity with Standard Deviation**
0 FTU	0.31 ± 0.02 FTU
1 FTU	1.03 ± 0.11 FTU
2 FTU	2.11 ± 0.22 FTU
6 FTU	5.99 ± 0.66 FTU
(**b**) Experimental target distances.	
**Set**	**Target Distances with Uncertainty**
Close	43, 48, 53, 58, 63 cm ± 1.5‰
Medium	73, 83, 93, 103 cm ± 1.5‰
Far	140, 170, 200 cm ± 1.5‰
(**c**) Experimental lighting settings.	
**Light Settings**	
25, 50, 75, 100%	

**Table 7 sensors-23-06575-t007:** Stereo depth accuracy and operation limits; dash “-” denotes no valid measurement. Δ denotes mean deviation from ground truth distance, shown with ±standard devation.

Target Dist.	Δ at 0.3 FTU	Δ at 1.0 FTU	Δ at 1.4 FTU	Δ at 2.1 FTU	Δ at 6.0 FTU
43.00 cm ± 1.5‰	11.30 (1.16) cm	11.80 (2.12) cm	11.76 (1.38) cm	11.58 (1.00) cm	12.12 (0.68) cm
48.00 cm ± 1.5‰	12.18 (1.27) cm	12.62 (1.17) cm	12.48 (0.93) cm	12.22 (0.90) cm	12.72 (0.91) cm
53.00 cm ± 1.5‰	12.42 (1.82) cm	12.78 (1.09) cm	12.70 (1.03) cm	12.64 (1.08) cm	13.34 (1.38) cm
58.00 cm ± 1.5‰	13.16 (1.41) cm	13.10 (1.33) cm	13.22 (1.11) cm	13.24 (1.30) cm	13.74 (1.60) cm
63.00 cm ± 1.5‰	13.36 (1.43) cm	13.74 (1.58) cm	13.64 (1.34) cm	13.64 (1.37) cm	13.92 (2.13) cm
73.00 cm ± 1.5‰	13.22 (11.08) cm	13.84 (1.87) cm	14.44 (2.65) cm	14.06 (1.77) cm	-
83.00 cm ± 1.5‰	12.58 (11.68) cm	13.64 (2.48) cm	13.54 (2.28) cm	13.44 (4.15) cm	-
93.00 cm ± 1.5‰	13.00 (2.77) cm	13.00 (2.81) cm	13.56 (3.06) cm	12.80 (2.74) cm	-
103.00 cm ± 1.5‰	10.38 (3.02) cm	11.12 (3.11) cm	10.40 (3.28) cm	11.14 (3.04) cm	-
140.00 cm ± 1.5‰	1.22 (6.47) cm	3.52 (5.76) cm	−0.32 (7.01) cm	0.46 (9.62) cm	-
170.00 cm ± 1.5‰	−10.98 (8.41) cm	−10.58 (9.34) cm	−13.82 (14.72) cm	−15.52 (19.36) cm	-
200.00 cm ± 1.5‰	−36.20 (23.68) cm	−33.30 (14.78) cm	-	-	-

**Table 8 sensors-23-06575-t008:** Laser triangulation accuracy and operation limits; dash “-” denotes no valid measurement. Δ denotes mean deviation from ground truth distance, shown with ±standard devation.

Target Dist.	Δ at 0.3 FTU	Δ at 1.0 FTU	Δ at 2.1 FTU	Δ at 6.0 FTU
43.00 cm ± 1.5‰	0.59 (0.70) cm	0.96 (0.73) cm	0.54 (0.30) cm	0.90 (0.22) cm
48.00 cm ± 1.5‰	0.33 (0.64) cm	0.49 (0.38) cm	0.61 (0.53) cm	-
53.00 cm ± 1.5‰	0.29 (0.89) cm	0.27 (0.42) cm	0.79 (1.19) cm	-
58.00 cm ± 1.5‰	0.20 (1.11) cm	0.28 (0.65) cm	0.21 (0.44) cm	-
63.00 cm ± 1.5‰	−0.05 (1.28) cm	−0.09 (0.40) cm	0.01 (0.84) cm	-
73.00 cm ± 1.5‰	−0.51 (1.30) cm	−0.62 (0.42) cm	−0.38 (3.11) cm	-
83.00 cm ± 1.5‰	−0.87 (1.14) cm	−1.00 (0.63) cm	−0.62 (0.49) cm	-
93.00 cm ± 1.5‰	−1.50 (0.73) cm	−1.40 (0.52) cm	−0.73 (0.71) cm	-
103.00 cm ± 1.5‰	−1.21 (2.00) cm	−1.54 (0.90) cm	−1.39 (0.98) cm	-
140.00 cm ± 1.5‰	0.42 (2.61) cm	-	-	-
170.00 cm ± 1.5‰	1.24 (2.79) cm	-	-	-
200.00 cm ± 1.5‰	2.64 (3.41) cm	-	-	-

**Table 9 sensors-23-06575-t009:** Imaging sonar accuracy and operation limits. Δ denotes mean deviation from ground truth distance, shown with ±standard devation.

Target Dist.	Δ at 0.3 FTU	Δ at 1.0 FTU	Δ at 1.4 FTU	Δ at 2.1 FTU	Δ at 6.0 FTU
43.00 cm ± 1.5‰	1.61 (2.18) cm	2.06 (2.10) cm	1.92 (2.41) cm	1.81 (2.05) cm	2.16 (2.41) cm
48.00 cm ± 1.5‰	1.80 (2.30) cm	2.36 (2.33) cm	1.81 (2.57) cm	2.10 (2.25) cm	1.97 (2.28) cm
53.00 cm ± 1.5‰	2.07 (2.41) cm	2.26 (2.30) cm	1.86 (2.82) cm	2.08 (2.17) cm	2.33 (2.28) cm
58.00 cm ± 1.5‰	2.18 (2.74) cm	2.40 (2.43) cm	2.14 (2.86) cm	2.29 (2.53) cm	2.62 (2.46) cm
63.00 cm ± 1.5‰	2.14 (2.55) cm	2.70 (2.10) cm	2.42 (2.90) cm	2.59 (2.34) cm	2.67 (2.00) cm
73.00 cm ± 1.5‰	2.21 (3.26) cm	2.74 (2.68) cm	2.51 (3.16) cm	2.47 (3.51) cm	3.08 (2.47) cm
83.00 cm ± 1.5‰	2.68 (3.92) cm	2.63 (3.13) cm	2.76 (4.60) cm	2.44 (3.94) cm	3.50 (3.82) cm
93.00 cm ± 1.5‰	2.51 (4.73) cm	2.83 (4.14) cm	3.24 (5.17) cm	2.49 (4.28) cm	2.97 (4.29) cm
103.00 cm ± 1.5‰	2.74 (5.13) cm	2.88 (4.47) cm	2.94 (5.54) cm	2.53 (4.70) cm	3.38 (4.57) cm
140.00 cm ± 1.5‰	5.94 (6.58) cm	4.25 (4.58) cm	2.23 (6.06) cm	2.42 (6.02) cm	3.06 (6.24) cm
170.00 cm ± 1.5‰	2.90 (5.68) cm	3.84 (4.28) cm	2.50 (5.58) cm	2.75 (5.32) cm	3.59 (5.52) cm
200.00 cm ± 1.5‰	3.06 (4.03) cm	4.05 (4.19) cm	2.38 (3.53) cm	2.81 (3.57) cm	3.57 (3.26) cm

## Data Availability

The data presented in this study are available on request from the authors.

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
