# Peer review of "Commercial Optical and Acoustic Sensor Performances under Varying Turbidity, Illumination, and Target Distances"

_sensors, 2023, doi:10.3390/s23146575_

Round 1
Reviewer 1 Report
In this paper, the authors quantify and compare the operational characteristics and environmental effects from turbidity and illumination on two sensors and an additional augmented optical method; a high-frequency forward-looking inspection sonar, a stereo camera with built-in stereo depth estimation, and a color imaging, where a laser has been added for distance triangulation. It is shown that optical stereo depth estimation and laser triangulation operate satisfactorily at low and medium turbidites, while acoustic measurements are almost completely unaffected.
It is quite interesting work. Here are two suggestions for the authors.
1. Try to reorganize the Introduction chapter. Provide some background in chapter of Introduction and move the related work to another new chapter.
2. Some text overlaps with Figures in the paper.
Author Response
Thank you very much for your comments
1) The introduction chapter has been reworked and the related works moved to a new section with some additions
2) The offending overlapping figure has been addressed
Reviewer 2 Report
The paper considers interesting and actual problem to design a calibration method for the underwater refraction model combining two refraction coplanar constraints with the aim of rapid calibration, which can improve the continuous operation capability of underwater 3D reconstruction system; as well as cross-validation of application for acoustic sonar, stereo camera depth estimation and laser triangulation.
Such a class of practical tasks is important for modern automated devices, first of all in autonomous target tracking by underwater robots. The paper has a good approach, and possesses a kind of formal structure in problem consideration. It should be of potential interest for readers. However, for possible publication it needs a revision.
As the guide for authors I can mention some points which should be reconsidered for better reader´s understanding of authors’ contribution:
1.Not sufficient related works analysis and too superficial. Authors are only concentrated on arbitrary selected methods, meanwhile some precise 3D measurement methods are already proposed in general theory of Machine Vision (for example, O. Y. Sergiyenko and V. V. Tyrsa, "3D Optical Machine Vision Sensors With Intelligent Data Management for Robotic Swarm Navigation Improvement," in IEEE Sensors Journal, vol. 21, no. 10, pp. 11262-11274, May, 2021 or in Book “Optoelectronic devices in Robotic Systems”. Edited by Oleg Sergiyenko. Editorial: Springer, October, 2022, 347p. ISBN: 978-3-031-09791-1. https://doi.org/10.1007/978-3-031-09791-1). I guess, in introduction must be considered and cross-compared more variety of methods for 3D points data acquisition.
2. Authors are considering on the optical scheme of Fig.1 so called “static laser triangulation”. However, the more versatile analysis can be reached comparing it with rotational dynamic laser triangulation.
3. The optical density and light transmissivity in underwater applications is more dependent on the application depth. Author should define in the text on what depth range their calibration is valid without modifications.
4. In my opinion, for better results convincing it is expedient summarize the results of all 3 applied methods on the same graph, preferably in homogenized conditions medium/target.
5. Table 8 on the page 12 of 22 and Table 9 on the page 14 of 22: should be useful the detailed explanation of Delta estimation methodology.
Assuming my conclusion, I can note that after careful revision and corresponding explanations the paper can be considered for publication.
Author Response
Thank you very much for your comments
1) Further references have been added to the reworked introduction and related works section
2) A note on future work with modulated and rotating lasers has been added to the concluding remarks
3) In doubt about the meaning for this point:
to the operating depth wrt. the water surface or
the measurement range of the investigated sensors?
Have currently addressed 3) in the first sense.
4) Due to the the many experimental parameters, our attempts at a combined figure did not yield a satisfactory overview (too cluttered).
5) Further reference to the meaning of delta and the calculating code used has been added.
In hope that this addresses your comments; thank you again.
Reviewer 3 Report
This manuscript describes very well the studied system. The details of the proposed stand and electronic system are presented. Also, the experimental tests were carried out to evaluate the performance of the method presented. The presented work is technically interesting.
I recommend more attention to the figures (figure 2 is too big and it is overall the text - line 118).
Author Response
Thank you very much for your comments
1) The offending overlapping figure has been adressed
Reviewer 4 Report
The comments are attached in PDF.

minor editing
Author Response
Thank you very much for your comments
1) The numerical upper bound for the error has been added in the abstract
2) The introduction has been reworked and a new related works chapter added
3) A few more references added
4) Grammar corrected, missing word was correct
5) Added further explanation by the reference to the figure 1
6) The offending figure has been adressed and updated
7) Figures have been reviewed for overlaps
8) Explanations added to abbreviations
9) Some more explanation added to section referencing Figure 5
10) Some more explanation added to section referencing Figure 6
11) Small grammatical corrections
12) Small grammatical corrections
Thank you once again.